# Biochemical and Hematological Correlates of Elevated Homocysteine in National Surveys and a Longitudinal Study of Urban Adults

**DOI:** 10.3390/nu12040950

**Published:** 2020-03-30

**Authors:** May A. Beydoun, Hind A. Beydoun, Peter H. MacIver, Sharmin Hossain, Jose A. Canas, Michele K. Evans, Alan B. Zonderman

**Affiliations:** 1Laboratory of Epidemiology and Population Sciences, NIA/NIH/IRP, Baltimore, MD 21224, USA; 2Department of Research Programs, Fort Belvoir Community Hospital, Fort Belvoir, VA 22060, USA; 3Department of Psychology, University of Maryland Baltimore County, Catonsville, MD 21228, USA; 4Department of Pediatrics, Johns Hopkins Medical Institutions, Saint Petersburg, FL 33701, USA

**Keywords:** homocysteine, hematological indices, biochemical indices, inflammation, predictive models, aging

## Abstract

Elevated blood homocysteine (Hcy) among middle-aged adults can increase age-related disease risk, possibly through other biochemical and hematological markers. We selected markers for hyperhomocysteinemia among middle-aged adults, studied time-dependent Hcy-marker associations and computed highly predictive indices of hyperhomocysteinemia, with cross-sectional and longitudinal validations. We used data from the National Health and Nutrition Examination Survey (NHANES III, phase 2, *n_max_* = 4000), the NHANES 1999–2006 (*n_max_* = 10,151) and pooled NHANES (cross-sectional validation). Longitudinal validation consisted of mixed-effects linear regression models (Hcy predicting markers’ annual rates of change), applied to the Healthy Aging in Neighborhoods of Diversity Across the Life Span (HANDLS, *n* = 227–244 participants, k = 2.4 repeats/participant, Age_base_: 30–65 years) data. Machine learning detected nine independent markers for Hcy > 14 µmol/L (NHANES III, phase 2): older age; lower folate and B-12 status; higher serum levels of creatinine, uric acid, alkaline phosphatase, and cotinine; mean cell hemoglobin and red cell distribution widths (RDW); results replicated in the 1999–2006 NHANES [AUC = 0.60–0.80]. Indices combining binary markers increased elevated Hcy odds by 6.9–7.5-fold. In HANDLS, first-visit Hcy predicted annual increase in creatinine, RDW and alkaline phosphatase, with third-visit index (2013–2018) directly predicting Hcy (2004–2009). We provide evidence of the internal and external validity of indices composed of several biomarkers that are strongly associated with elevated Hcy.

## 1. Introduction

Homocysteine (Hcy) is a sulfur amino acid involved in the remethylation and transsulfuration metabolic pathways, with the first requiring folate and vitamin B-12 as coenzymes, while the second depends on a form of vitamin B6, pyridoxal 5-phosphate [1]. Epidemiological evidence indicates that elevated Hcy (>14 µmol/L) can increase risk for cardiovascular and cerebrovascular disease and may double the risk for Alzheimer’s Disease (AD) [1,2,3,4,5]. Although biologically plausible, the causal nature of the AD–Hcy association remains a subject of debate. However, predicting Hcy from more commonly measured biochemical and hematological markers and creating a highly predictive index of elevated Hcy can be used in future cohort studies [6,7].

During one-carbon metabolism (OCM) cycles, one key enzymatic reaction involves re-methylation of Hcy, whereby a methyl group is acquired from N-5-methyl-tetrahydrofolate (MTHF) or from betaine to form methionine. While the former reaction requires folate and vitamin B12, the latter does not [8]. Adenosine triphosphate (ATP) is then used to convert methionine to S-adenosylemethionine (SAM), a universal methyl donor utilized by various acceptors including nucleic acids, hormones and neurotransmitters [8]. A methyl donation by-product, S-adenosylhomocysteine (SAH), is further hydrolyzed to regenerate Hcy, starting a new cycle of methyl group transfer [8]. The transsulfuration pathway then catabolizes excess Hcy not required for methyl donation into cysteine, using a vitamin-B-6-dependent enzyme, and cysteine is later oxidized to taurine and inorganic sulfates or excreted in urine [8]. Serum folate and vitamin B-6 and B-12 levels are strong inverse predictors of elevated blood Hcy [9,10,11,12,13,14], as are genetic polymorphisms associated with the OCM, such as MTHFR C667T, associated with reduced methylene tetrahydrofolate reductase (MTHFR) enzymatic activity [15]. Nevertheless, unexplained variability can be ascribed to kidney disease, explaining a positive association between Hcy and serum creatinine [5,16,17,18,19]. It is worth noting that men with higher muscle mass have been shown to have higher levels of both Hcy and creatinine, particularly when compared to women, given that ~70% of daily SAM-dependent methylation reactions are to produce creatine [20,21]. Hcy has been positively associated with red cell distribution width (RDW) [22]; with increased serum cotinine, a measure of active or passive recent smoking [23]; and with increased liver enzyme levels [24,25]. Other unexplored biochemical and hematological markers may also be predictive of elevated Hcy which could be reflecting other risk factors for age-related disease such as cardiovascular and neurodegenerative disorders. Generally, there is a paucity of research in the following areas: (1) discovering the most predictive measures of elevated Hcy out of selected biochemical and hematological markers; (2) creating indices that can be used as surrogates of elevated Hcy in studies which do not measure Hcy per se; (3) enhancing understanding as to why elevated Hcy may increase the risk of certain age-related diseases, including AD, by discovering novel markers that are highly predictive of elevated Hcy.

Thus, no study to date has examined and compared potential biochemical and hematological predictors of Hcy among middle-aged adults in a systematic and exploratory manner, by combining machine learning and receiver operating characteristic (ROC) techniques. This novel approach can be applied in other future studies examining the predictors of other clinical mediators of disease. Thus, our present study aimed at selecting a comprehensive yet parsimonious predictive model of elevated Hcy among middle-aged adults, using biochemical and hematological data from the third and most recent (1999–2006) National Health Nutrition Examination Survey, a model cross-validated in a longitudinal study of urban adults, from which an index reflecting elevated Hcy was also validated.

## 2. Materials and Methods

### 2.1. Databases

#### 2.1.1. NHANES III, Phase 2 and 1999–2006

The National Health and Nutrition Examination Survey (NHANES) was conducted following guidelines laid down in the Declaration of Helsinki, and all procedures involving human subjects/patients were approved by the Institutional Review Board of the National Center for Health Statistics, Centers for Disease Control and Prevention (CDC). Written or verbal informed consent was obtained from all participants; verbal consent was witnessed and formally recorded [26].

NHANES consists of cross-sectional surveys providing nationally representative data on the health and nutritional status of the U.S. civilian population. Initiated in the 1970s by the National Center for Health Statistics (NCHS), CDC, earlier waves of NHANES collected data in non-continuous fashion. Since 1999, NHANES became a continuous survey. The sampling design is stratified and multistage-probability-clustered. It includes an in-home interview for demographic and basic health information completed by trained staff and a health examination in a mobile examination center (MEC), completed by physicians, medical/health technicians, and dietary and health interviewers [26]. Of interest are NHANES waves with complete blood Hcy data, namely NHANES III, phase 2 (1991–1994) [27] and the 1999–2006 wave [26]. Regulations for mandatory fortification of wheat flour with folic acid, currently in place in 53 countries, were implemented in the United States in 1998, adding 140 µg of folic acid per 100 g of enriched cereal grain product, and have been estimated to provide 100–200 µg of folic acid per day to women of childbearing age, ultimately reducing the incidence of neural tube defects [28]. This also resulted in a reduced prevalence of elevated Hcy over time and specifically between the two waves of NHANES used in this study [28].

NHANES specimen storage was consistent across waves. Upon arrival at the CDC or contract laboratories, the frozen specimens were sorted by vial type, and stored initially at -20 C. The refrigerated samples were stored at 4–8 C. Frozen specimens whose analysis might have been delayed were stored at -70 C or lower [29].

#### 2.1.2. HANDLS 2004–2018

HANDLS is an ongoing prospective cohort study initiated in 2004. It focuses primarily on disparities in the cardiovascular and cognitive health of a socioeconomically diverse sample of Whites and African Americans aged 30–65 yo at baseline and living in selected neighborhoods of Baltimore, Maryland. In brief, HANDLS used an area probability sampling strategy of thirteen neighborhoods, with details provided elsewhere [30]. Phase 1 of Visit 1 (2004–2009) consisted of screening followed by recruitment, household interviews, while phase 2 of Visit 1 (also 2004–2009) consisted of in-depth examinations in a mobile Medical Research Vehicle (MRV), including measurements of blood pressure; anthropometrics and a fasting blood draw were also collected at the follow-up visits [Visit 2: 2009–2013; Visit 3: 2013–2018]. Although blood Hcy was measured only in a small subset of Visit 1 participants (i.e., at baseline), all other available hematological and biochemical indices had three repeats at Visits 1, 2 and 3 (2004–2009, 2009–2013 and 2013–2018). All clinical laboratory indices were obtained from Quest Diagnostics (Chantilly, VA). Mean follow-up times between visits ranged between 6 months and 8 years, with an average of 4–5 years.

Participants provided written informed consent after reviewing a protocol booklet written in layman’s terms and watching a video detailing all procedures and future re-contacts. The HANDLS study was approved ethically by the Institutional Review Board of the National Institutes of Health, National Institute of Environmental Health Sciences (NIEHS/NIH).

### 2.2. Study Samples

We selected adults aged 30–65 years from the NHANES III (phase 2) and from the 1999–2006 waves. Similarly, by design, Visit 1 of HANDLS consisted of adults aged 30–65 years (Appendix A). In the NHANES III, phase 2, biomarkers with >20% missing data compared to the sub-sample with complete Hcy measures were excluded. Out of 15,283 participants from phase 2 of NHANES III, 8585 had complete Hcy data, of whom 4008 were in the age range of interest. Of those, 3709–4000 had complete data on up to 82 biochemical and hematological markers. Similarly, for the NHANES 1999–2006, of an initial 41,474 participants, completeness on Hcy data was found for *n* = 28,449, of whom 10,151 were aged 30–65 years and the final analytical sample ranged between 9991 and 10,151, after biomarkers were selected with machine learning methods using NHANES III, phase 2 data. Pooling data from NHANES III, phase 2 and NHANES 1999–2006, 14,739–14,829 provided complete data on Hcy and the selected biochemical and hematological indices, within the age range 30–65 years. Finally, out of 3720 HANDLS participants, only 245 individuals had complete data on Hcy measured during the first MRV visit. All these participants had the target Visit 1 age of 30–65 years, and the final analytic samples for longitudinal analysis ranged between 227 and 244 individuals with multiple repeats (up to 3, mean repeats/participants, k = 2.4), depending on adjustment levels.

### 2.3. Serum Homocysteine

In NHANES III, phase 2, serum Hcy was measured at the Jean Mayer USDA Human Nutrition Research Center on Aging, Tufts University, using the high-performance liquid chromatography method of Araki and Sako [31]. In the recent NHANES, serum Hcy was measured using “Abbott Homocysteine (HCY) assay”, a fully automated technique [32,33], as was Hcy in the HANDLS sub-cohort. In all datasets, elevated Hcy was defined as >14 μmol/L, a cut-point of 2.639057 on the Log_e_-transformed scale used to examine the association between elevated Hcy and AD in most previous studies [1,2,3,4,5].

### 2.4. Biochemical and Hematological Indices

Biochemical indices included nutritional biomarkers (e.g., folate, B-12, vitamin D, vitamin E, carotenoids, retinol, vitamin C, total calcium, iron, sodium, potassium etc.), metabolic parameters (e.g., serum insulin, glucose, cholesterol, triglycerides, creatinine, albumin, thyroid hormones, liver enzymes) and inflammatory markers (e.g., C-reactive proteins, Immunoglobulin G (IgG) against specific viruses and bacteria), and environmental indices of air pollution and smoking (e.g., blood lead and serum cotinine).

As stated earlier, lower serum folate and vitamin B-12 concentrations are among the highly predictive markers of elevated Hcy. In NHANES III, phase 2, serum folate and B-12 were measured using Bio-Rad Laboratories “Quantaphase Folate” radioassay kit [29,34], as was the case for more recent NHANES [35,36]. In HANDLS, these two measures were determined using enzyme immunoassay by Quest Diagnostics, Chantilly, VA [37], at Visits 1 through 3. Hematological indices consisted of markers of blood cell counts and characteristics (Appendix A).

### 2.5. Covariates

In all NHANES predictive models, the following covariates were considered: age, sex, race/ethnicity (1: NH white, 2: NH black, 3: Mexican American, 4: other Hispanic, 5: others), poverty status (0: >125% of poverty income ratio; 1: ≤125% of poverty income ratio), rural vs. urban area of residence, and region (Northeast, Midwest, South and West). Given that the last two factors are fixed in HANDLS (urban, Northeast: Baltimore city), only age, sex, race (African American vs. Whites) and poverty status were included, using a similar cut-point ≤125% of the federal poverty line.

### 2.6. Data Handling and Statistical Analysis

All analyses were conducted using Stata release 16.0 [38]. We first describe study characteristics [covariates (all datasets, waves); Log_e_-transformed Hcy; Log_e_-transformed biochemical and hematological indices (NHANES III, phase 2)], overall and by categorical Hcy (≤14 μmol/L vs. >14 μmol/L). Differences in means and proportions across these categories were tested using a design-based F-test accounting for sampling design complexity. Beyond this descriptive step, a multi-stage approach was implemented to select the key predictors of elevated Hcy in NHANES III, phase 2, validate them against the most recent NHANES, and cross-validate those predictors in a longitudinal study of urban adults (HANDLS). A flow diagram is used to summarize this approach (Figure 1).

To select predictive biomarkers of continuous Hcy (Log_e_-transformed, z-scored), a statistical learning method known as least absolute shrinkage and selection operator (LASSO) was used. LASSO is a covariate selection methodology that is superior to both generalized linear models without covariate selection and the usually applied stepwise or backward elimination process [39]. In fact, stepwise selection is often trapped into a local optimal solution and backward elimination can be time-consuming [39]. The LASSO, which does not ignore stochastic errors, is defined as follows:(1)$$\beta (\mathrm{lasso})=\arg min_{\beta }{\left\|y-\sum_{j=1}^{p}x_{j}\beta_{j}\right\|}^{2}+\lambda \sum_{j=1}^{p}\left|\beta_{j}\right|$$
with λ being a non-negative regularization parameter [39]. The second term of the equation, termed the “l1 penalty”, is a key portion of this equation, ensuring the success of the LASSO method of covariate selection [39]. In fact, this method was shown to discover the right sparse representation of the model, given certain conditions [39]. More recently, several related methods have been developed and validated against each other, with an adaptive LASSO giving more consistent findings, particularly when compared with the non-negative garotte [39].

In our predictions, we used this convex optimization technique with an l_1_ constraint, known as adaptive LASSO, as the main method to select the final linear regression model for prediction of Log_e_-transformed Hcy with Log_e_-transformed and z-scored biomarkers and socio-demographic factors, with the latter being force entered into all models. The model was trained on a random half sample of the total population (among the target age group: 30–65 years, sorting the sample by individual ID and fixing a random seed) and validated against the other half sample to check robustness of findings, by comparing R^2^ between samples. Adaptive LASSO robustness is then compared to that of cross-validation (cvLASSO) and minimal Bayesian information criterion (minBIC) LASSO, and non-zero parameters were presented for each method. This parsimonious model, with Log_e_-transformed Hcy as an outcome, was then run on the entire population accounting for survey design complexity (i.e., svy: reg) as a starting point for further backward elimination. Thus, beyond that point, additional terms were eliminated at a type I error of 0.10. This final model was applied to the binary outcome of elevated blood Hcy (>14 µmol/L), using svy: logit, and further backward elimination was carried out to obtain a short list of independent predictors for elevated Hcy. As a sensitivity analysis, additional markers identified with adaptive LASSO logistic regression on the same half-sample as for the adaptive LASSO linear regression were included in the reduced model to test their predictive value.

In the full NHANES III, phase 2, the selected Log_e_-transformed biomarkers and continuous socio-demographic variables (e.g., age) in the final models were then entered into a series of ROC analyses to determine the most appropriate cut-point, which would have the largest sensitivity and specificity in predicting elevated Hcy (>14 μmol/L). Sensitivity (proportion of true positives, i.e., proportion of cases correctly identified as meeting the conditions of elevated Hcy) and specificity (proportion of true negatives, i.e., proportion of non-cases correctly identified as not meeting elevated Hcy) were calculated to evaluate accuracy of selected biochemical and hematological markers in depicting elevated Hcy, creating a series of ROC curves [40,41]. The ROC curve is a graphical plot of sensitivity vs. (1 - specificity) for a binary classifier system as its discrimination threshold is varied. The area underneath of each ROC curve (AUC), a measure that is independent of classifier cut-points, can range between 0 and 1 and be computed with its 95% confidence interval (95% CI).

A Log_e_-transformed biomarker positively associated with elevated Hcy would yield an AUC between 0.5 and 1.0. An area of 0.70, for instance, has the following interpretation: if we randomly select a biomarker from the Hcy^+^ and Hcy^−^ groups, the value of that biomarker will be greater in the Hcy^+^ group than in the Hcy^−^ group, 70% of the time. The ROC curves and their associated AUC are presented, with biomarkers inversely linked to elevated Hcy having their values inverted (i.e., multiplying them by −1). Subsequently, biomarkers retained in the final logistic regression model (selected with machine learning and backward elimination), were further pruned out when ROC AUC was <0.55. Thus, only biomarkers with AUC ≥ 0.55 were retained and their AUC and optimal cut-points presented for NHANES III, phase 2, validated against NHANES 1999–2006 and presented for the pooled NHANES.

An index combining all selected binary biomarkers was computed with a potential to range between 0 and *m’* [number of selected biomarkers with positive criterion: ≥optimal cut-point]. The final index summing categorical biomarkers was computed using revised cut-points from pooled NHANES data. Two ordinal indices were obtained, namely, Index I summing all selected binary biomarkers reflecting elevated Hcy that were available in all selected NHANES waves, and Index II, sub-set of Index I, using only commonly measured biomarkers available in the HANDLS study. Similarly, a ROC analysis was conducted on the pooled NHANES to determine the optimal cut-point for Indices I and II. To determine potential use of those biomarkers as surrogates for elevated Hcy in other large epidemiological studies, a logistic regression model was conducted with each Index and with Index I components entered simultaneously in the pooled NHANES data.

Importantly, the cross-sectional and longitudinal associations of Visit 1 Hcy (2004–2009) on repeated measures of the biochemical and hematological markers (Visits 1 through 3: 2004–2018) that were selected for NHANES (continuous) were tested in a sub-set of HANDLS using multiple mixed-effects linear regression analysis (See Appendix A). All models were adjusted for Visit 1 age (Model 1), with further adjustment for sex, race and poverty status applied to Model 2, while the full model (Model 3) additionally adjusted for all remaining biomarkers. Finally, Index II, computed using the NHANES cut-point, was computed at Visit 3 and correlated with Hcy at Visit 1 of HANDLS using Pearson’s correlation and locally weighted regression (LOWESS) smoother, to assess external validity of the association. Type I error was set at 0.05 with *p* < 0.10 considered as a trend or tendency towards an association.

## 3. Results

Table 1 present study sample characteristic distributions, namely Hcy and socio-demographic factors for both NHANES waves and for HANDLS Visit 1, as well as biochemical and hematological correlates for NHANES III, phase 2, overall and stratified by Hcy status. In all samples, age and sex (men vs. women) were consistently associated with elevated Hcy, while poverty was directly associated with elevated Hcy only in NHANES 1999–2006, and both race/ethnicity and poverty status trended towards an association with this binary outcome in the HANDLS sub-cohort. Numerous biochemical and hematological indices were significantly associated with elevated Hcy, including serum cotinine, mean cell hemoglobin (MCH), RDW, blood lead, serum uric acid (SUA), serum creatinine, serum alkaline phosphatase, while others were inversely linked to elevated Hcy, namely serum vitamin E, most serum carotenoids, serum retinyl esters, serum folate and vitamin B-12 (*p* < 0.05). It is worth noting that those associations are crude, not adjusted for socio-demographic factors such as age and sex. Among the known predictive factors, serum folate was shown to have an unadjusted mean of 7.6 in the elevated Hcy group compared to 13.6 in the normal group, suggesting a strong inverse relationship.

Cross-validation (cv), adaptive and minBIC LASSO results are presented in Appendix A, using a random half sample of NHANES III, phase 2 and allowing for replication by sorting the sample by ID and setting a random seed to fixed value. Our findings indicated that Log_e_-transformed Hcy was associated with a number of biochemical and hematological indices, forcing adjustment for socio-demographic factors, most of which were shown to associate with elevated Hcy in Table 1. The result of the adaptive LASSO (initial model of choice) followed by backward elimination process is shown in Appendix A for both continuous and binary Hcy outcomes, while accounting for survey design complexity. In the reduced logistic regression model, the finally selected predictors included: serum folate (−), creatinine (+), age (+), serum vitamin B-12 (−), aspartate aminotransferase (+), alanine aminotransferase (−), SUA (+), mean cell hemoglobin, MCH (+), serum albumin (+), serum vitamin C (+), RDW (+), alkaline phosphatase (+), retinyl esters (−) and serum cotinine (+). Additional control for five markers identified by LASSO logistic (adaptive method, Appendix A) and not by the LASSO linear (adaptive method) did not alter this finding. Of those, only 10 markers survived the selection criteria of AUC > 0.55, and one (retinyl esters) was excluded due to missingness in recent waves. The remaining nine components were retained, with related ROC curves; estimated optimal cut-points for highest sensitivity/specificity are presented in Appendix A. For NHANES 1999–2006, the ROC curves were similar for each of the nine components and cut-offs were comparable using the same criterion for elevated Hcy. The pooled NHANES similarly yielded a mid-range value of cut-points as indicated in Appendix A.

The nine-component Index I predicted elevated Hcy with an AUC of 0.798 (95% CI: 0.783,0.812) in the pooled NHANES data (Figure 2). Similarly, Index II, which excluded serum cotinine, thus including eight components, exhibited an AUC of 0.794; 95% CI: 0.780,0.809 (Figure 3), indicating that when two values of Indices I and II were chosen at random, the lower value corresponded to a ≤14 value of Hcy and the higher value to >14 value of Hcy, ~80% of the time, suggesting a high predictive value of both indices for elevated Hcy. Optimal cut-point for both indices was 5. For each binary index (≥5 vs. <5) and in the pooled NHANES, (Table 2), the adjusted odds of elevated Hcy were increased 6.9–7.4-fold. Each of the nine binary components of Index I, when included into the model, simultaneously predicted elevated Hcy, independently increasing the odds by >24% (higher RDW) up to 3.5-fold (lower serum folate).

In the HANDLS sub-cohort analyses (Table 3), we found that Visit 1 Hcy was cross-sectionally associated, after multivariable adjustment, with lower serum folate, higher serum creatinine, lower serum vitamin B-12, and increases levels of SUA and alkaline phosphatase. Longitudinally, the multivariable adjusted mixed-effects regression model indicated that Visit 1 Hcy was linked to faster rate of increase in serum creatinine and serum vitamin B -12, with a trend towards a direct association (*p* < 0.10) with rates of increase in RDW and alkaline phosphatase. Finally, Visit tHcy (Log_e_-transformed) was found to be moderately and positively correlated with Visit 3 Index II, computed using pooled NHANES cut-points, with a Pearson’s correlation *r* = +0.34 (*n* = 81). The smoothed positive association is depicted in Figure 4, indicating a linear relationship for the range of the data.

## 4. Discussion

Here we present data among middle-aged adults, validating a predictive index for Hcy > 14 µmol/L derived from independent biochemical and hematological correlates using modern techniques. The study uncovered up to nine independent predictors for elevated Hcy, some of which have been found to be correlated with each other (e.g., serum folate, B-12, MCH, and RDW) in previous studies as well as with hyperhomocysteinemia. Both adjusted and unadjusted associations indicated that serum folate was the most predictive factor that was inversely related to elevated Hcy. Our findings of inverse associations of serum folate and cobalamin with elevated Hcy concentrations conform with earlier research using national data from pre-folate and post-folate fortification eras [17,19,42]. In fact, the two previous NHANES studies (III, phase 2:1991–1994 and 1999–2004) reported comparable findings, despite examining pre-selected factors, rather than exploring all available biochemical and hematological markers. The NHANES III study concluded that serum creatinine and cobalamin concentrations showed the strongest and weakest association with blood Hcy, respectively [19]. Notably, folate and vitamin B-12 were inversely related to Hcy. Men had higher mean Hcy than women, along with lower concentrations of serum folate, red blood cell (RBC) folate, and serum vitamin B12. The NHANES 1999–2004 study concluded that blood Hcy concentration was ~9.7% higher in men vs. woman [17], and was directly related to systolic blood pressure, serum creatinine, and serum cotinine, while being inversely correlated with serum folate levels, RBC folate, and serum vitamin B-12, and positively correlated with methylmalonic acid (MMA) concentration [17]. While those associations were largely replicated in our study, age and not sex was retained in the model upon backward elimination. Moreover, our study excluded other lifestyle or health-related factors, selecting only biochemical and hematological indices measured in NHANES III, phase 2 (1991–1994) along with Hcy. Consequently, MMA was excluded due to its unavailability in the NHANES III phase 2 data from among the 82 selected biomarkers.

Among retained correlates, SUA was strongly related to Hcy. In a retrospective cohort study (*n* = 16,477, age: 20–80 years), elevated SUA was previously directly associated with hyperhomocysteinemia, whereby the fully adjusted association remained significant only among men (OR = 1.5; 95% CI, 1.3,1.7; *p* < 0.001) [43]. These results are comparable to our findings, whereby SUA ≥ 339 µmol/L was associated with a 1.67-fold increase in the odds of hyperhomocysteinemia (>14 µmol/L), with a 95% CI: 1.35,2.06.

Hyperuricemia can be modified with diet as well, including reduced alcohol, red meat and sugar consumption [44,45].

High MCH is commonly a sign of macrocytic anemia (i.e., enlarged RBCs) subsequent to folate or vitamin B-12 deficiency [46], though it may also result from liver diseases [46]. Thus, Hcy may well be a correlate of higher MCH, resulting from any or both vitamin deficiencies; and a higher Hcy may result in higher MCH over time. Nevertheless, our findings indicated that although an independent correlate of elevated Hcy, MCH was not among the strongest predictors based on ROC analyses. Furthermore, our longitudinal analyses did not indicate that baseline Hcy was associated with faster increase in MCH over time. Nevertheless, larger studies are needed to corroborate these findings. Elevated RDW, reflecting RBC size variability (i.e., anisocytosis), independently predicted chronic disease morbidity and mortality [47,48,49,50,51]. Unlike MCH, RDW was previously studied in relation to hyperhomocysteinemia [22,52], and was found to be directly related to elevated Hcy in one cross-sectional study of middle-aged Chinese adults, independently of age, neutrophil count, mean corpuscular volume, and hemoglobin [22]. Another larger cross-sectional study of 5554 adults (18–64 years), however, failed to detect this independent RDW-Hcy [52]. Our findings indicated that elevated MCH was in fact more strongly associated with elevated Hcy (OR = 1.60, 95% CI: 1.28,2.02) when compared with RDW (OR = 1.24, 95% CI:1.01,1.54). Thus, elevated Hcy may be a stronger marker of enlarged RBCs than of anisocytosis. Nevertheless, our longitudinal analysis has shown that Hcy in its continuous form predicted RDW to a greater extent than MCH, and was associated with a faster rate of increase in RDW over time.

Moreover, serum bone alkaline phosphatase, a marker of biliary inflammation and cholelithiasis [53], was shown to be up-regulated in vitamin B-6 deficiency [25,54]. As stated earlier, Hcy was previously inversely corelated with vitamin B-6 status [9,10,11,12,13,14], as the latter is directly involved in OCM. Our study is to date the first to show that higher blood Hcy is associated with elevated alkaline phosphatase, both cross-sectionally and longitudinally. Thus, although the main modifiers of Hcy are B-vitamins, particularly folate and vitamin B-12, liver enzymes are correlates of Hcy that can be modulated with reduced alcohol consumption [55], and alkaline phosphatase in particular is a key mediator in the reported association between Hcy and reduction in bone mineral density associated with osteoporosis among postmenopausal women [56,57]. Thus, Hcy may be merely a marker of certain health outcomes, while liver enzyme elevations can act as the main causal pathway.

Several studies have indicated that smokers had more elevated blood Hcy than non-smokers, independently of other factors, while having lower serum levels of folate and vitamin B-12 [23]. Among self-reported never smokers >20 years of age [NHANES III, *n* = 3232], serum cotinine quartiles were independently and linearly associated with blood Hcy, as were age, being male, being non-white, and having lower sum folate or serum B-12 [58]. Serum cotinine was also among key predictors of elevated Hcy in studies examining multiple correlates in earlier and more recent NHANES [17,19]. Thus, stopping cigarette smoking may have an effect of reducing the risk of elevated Hcy. Nevertheless, controlled randomized trials are needed to ascertain a causal association. It is worth noting that both serum cotinine and urinary cadmium were linked to recent smoking [59,60].

Several notable study strengths include the novel coupled use of machine learning and ROC analyses to select independent predictors for elevated Hcy and subsequently create combined indices and conduct multivariable regression models. The initial analysis screened over 82 biochemical and hematological biomarkers, and our confidence in the predictive indices was enhanced by validation between cross-sectional national data and an independent longitudinal study of urban adults. The LASSO linear model was used to obtain a first set of predictors for continuous Log_e_-transformed Hcy, which were then applied to the binary outcome, given that the 14 µmol/L cutoff to define elevated Hcy might be considered arbitrary for some health outcomes, aside from AD. Thus, our goal was to limit the set of markers to those that independently predicted Hcy, both in its continuous and categorical form.

Among the limitations, the threshold used for Hcy of 14 µmol/L in our study, while being used by others previously, may be sub-optimal in some samples, given their different levels of mean Hcy, particularly given the decreasing prevalence rates between pre- and post-folate fortification. Thus, even though cutoffs for predictors were comparable at optimal sensitivity and specificity between NHANES waves, the predictive values may have differed between those two waves, with expected higher positive predictive value at higher prevalence of elevated Hcy (i.e., NHANES III, phase 2) and vice versa for the negative predictive value. Second, Hcy measurement, while comparable between waves, used different techniques between NHANES III, phase 2 and the more recent NHANES, potentially affecting the validity of the cut-point used between those two waves. Nevertheless, given that comparable biomarker optimal cut-points were obtained between NHANES waves through ROC analyses for Hcy > 14 µmol/L, measurement errors ascribed to differential use of techniques (HPLC vs. immunoassay) was assumed minimal. Attempts to calibrate those two methods are needed in future studies with repeat measures using both methods. Finally, our key findings and the indices derived from ROC analyses may be applicable only at mid-life, a time window whereby cardiovascular and neurodegenerative diseases can be prevented through Hcy-reducing interventions. Nevertheless, future studies should examine those relationships and validate those indices among older adults aged ≥ 65 years.

In sum, we provide evidence of internal and external validity of indices composed of several biochemical and hematological markers that are strongly associated with elevated Hcy, which may be used as proxies in future longitudinal studies. Components of those indices that are amenable to intervention (e.g., folate and B-12 supplementation, alcohol consumption which affects both liver enzymes and uric acid, cigarette smoking) should also be studied as alternative pathways for which elevated Hcy can affect cardiovascular and neurodegenerative disease trajectories.

## Figures and Tables

**Figure 1 nutrients-12-00950-f001:**
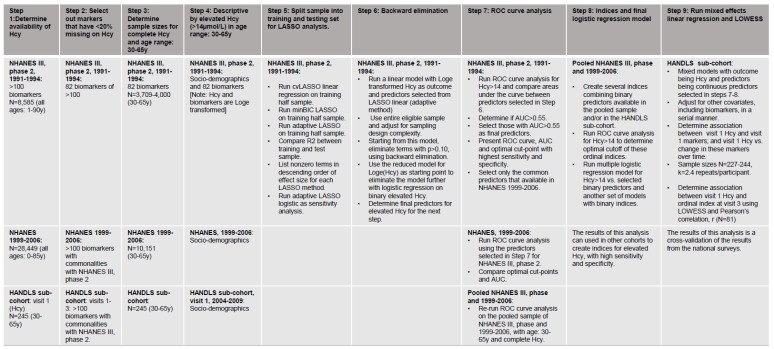
Flow diagram of predictive modeling using LASSO, ROC curves and multivariable regression modeling. Abbreviations: cvLASSO = cross-validation LASSO; LASSO = least absolute shrinkage and selection operator; LOWESS = Locally weighted regression; HANDLS = Healthy Aging in Neighborhoods of Diversity Across the Life Span; minBIC LASSO = minimum Bayesian information criterion LASSO; NHANES = National Health and Nutrition Examination Surveys; R2 = coefficient of determination.

**Figure 2 nutrients-12-00950-f002:**
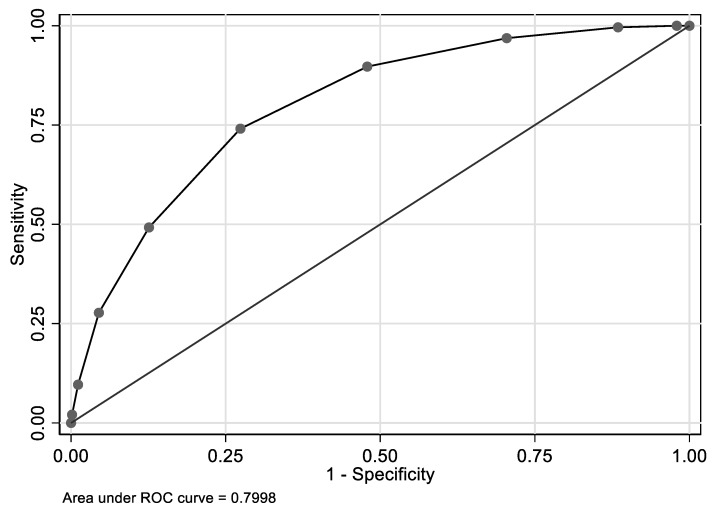
Nine-marker index (Index I) and its predictive value of elevated Hcy: ROC curves for pooled NHANES ^1^. ^1^
*n* = 13,822; optimal cut-point was at 5; AUC = 0.799, 95% CI: 0.785,0.814. Index I included binary biomarkers of elevated Hcy selected using LASSO and backward elimination. The full list of the nine components of Index I are: Age, serum folate, serum vitamin B-12, serum creatinine, red cell distribution width, mean cell hemoglobin, serum cotinine, serum uric acid and alkaline phosphatase. Cut-points for individual components are: serum folate, Log_e_, in nmol/L, <2.83; serum creatinine, Log_e_, in µmol/L ≥4.481; older age, in years, ≥49; serum vitamin B-12, Log_e_, in pmol/L, <5.74; mean cell hemoglobin, Log_e_, in *pg*, ≥3.422; red cell distribution width, Log_e_, in %, ≥2.553; serum uric acid, Log_e_, in µmol/L, ≥5.826; serum alkaline phosphatase, Log_e_, in µmol/L, ≥4.356 U/L; serum cotinine, Log_e_, in ng/mL, −0.579.

**Figure 3 nutrients-12-00950-f003:**
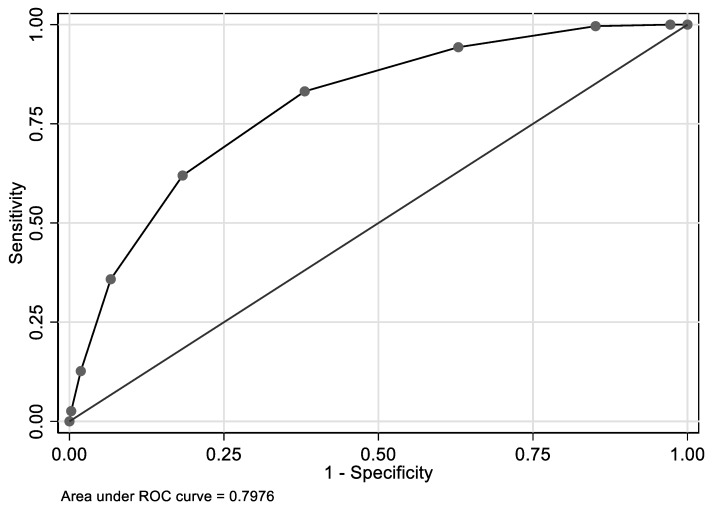
Eight-marker index (Index II) and its predictive value of elevated Hcy: pooled NHANES ^1^. ^1^
*n* = 13,920; optimal cut-point at 5; AUC = 0.798; 95% CI: 0.783,0.812. The index consisted of a summation of all binary Index I biomarkers (See Figure 2 footnotes for cut-points), excluding blood cotinine, which was not available in HANDLS.

**Figure 4 nutrients-12-00950-f004:**
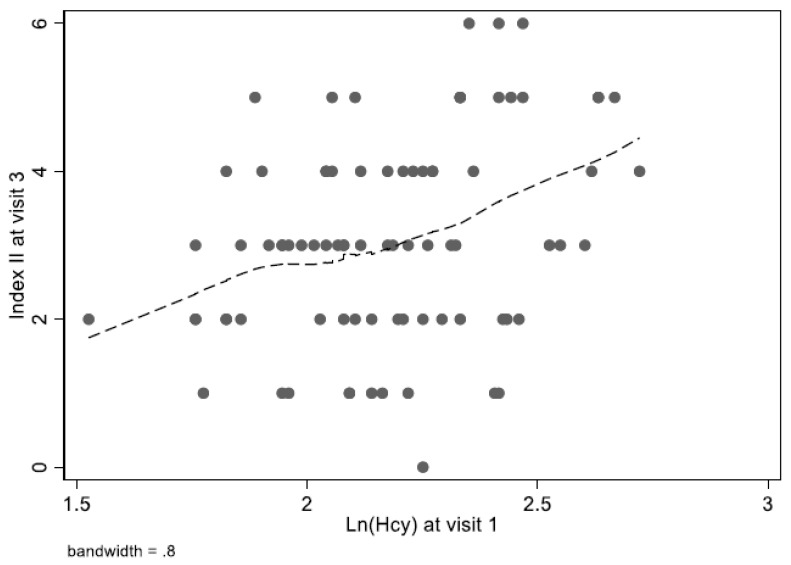
Lowess smoother of total Visit 1 homocysteine vs. Visit 3 Index II (*r* = 0.34, *n* = 81): HANDLS 2004–2009 (Hcy) and 2013–2018 (Index II) ^1^. Abbreviations: HANDLS = Healthy Aging in Neighborhoods of Diversity Across the Life Span: Hcy = Homocysteine; NHANES = National Health and Nutrition Examination Surveys; ROC = Receiver Operating Characteristics analysis. ^1^ Blood total Hcy, in µmol/L, is Log_e_-transformed and uncategorized. Index II was computed using cut-points obtained from the pooled NHANES ROC analysis. See Table 2 footnotes for details. Index II may range from 0 to 8, and no cut-point was used in this analysis.

**Table 1 nutrients-12-00950-t001:** Participant characteristics distribution by hyperhomocysteinemia status: NHANES III, phase 2; NHANES 1999–2006 and HANDLS 2004–2018 ^1^.

	**Overall**				**Hcy ≤ 14 µmol/L**				**Hcy > 14 µmol/L**				P_Hcy_
	***n***	**Mean**	**%**	**SE**	***n***	**Mean**	**%**	**SE**	***n***	**Mean**	**%**	**SE**	
NHANES III, phase 2: 1991–94	4000		100.0	0.0	3663		91.5	0.7	337		8.4	0.7	
Age (y)	4000	44.3		0.42	3663	44.0		0.45	337	47.1		0.76	0.001
Sex, % men	4000		49.0	1.2	3663		47.8	1.2			62.8	4.9	0.008
Race/ethnicity	4000				3663				337				0.51
NH white			74.8	2.1			74.6	2.2			75.8	3.7	
NH black			11.1	1.0			10.9	1.1			13.0	1.9	
MA			5.2	0.7			5.4	0.7			3.7	0.7	
Other			8.8	1.6			9.0	1.6			7.6	2.6	
Poverty status													
PIR ≥ 125%	3714		14.0	1.9	3388		14.0	1.9	326		14.6	2.3	0.79
PIR < 125%													
Region	4000		18.9	2.2	3663		18.4	2.5	337		23.8	4.8	0.30
Northeast			22.6	4.4			22.8	4.5			20.7	4.4	
Midwest			36.0	7.3			35.8	7.4			36.9	7.6	
South			22.6	7.5			22.9	7.6			18.3	6.9	
West													
Urban/Rural	4000				3663				337				0.59
Urban			50.5	7.5			50.7	7.6			48.2	8.2	
Rural			49.6	7.5			49.3	7.6			51.8	8.2	
Hcy, Log_e_	4000	+2.17		0.01	3663	+2.10		0.01	337	+2.93		0.04	<0.001
**Selected biochemical and hematological indices, Log_e_**		**Mean,** **Log_e_**	**Mean,** **exp**	**SE,** **Log_e_**		**Mean,** **Log_e_**	**Mean,** **exp**	**SE,** **Log_e_**		**Mean,** **Log_e_**	**Mean,** **exp**	**SE,** **Log_e_**	
Serum cotinine, ng/mL	3966	+0.33	1.39	0.12	3630	+0.17	1.190	0.11	336	+2.08	8.0	0.27	<0.001
Serum vitamin D, nmol/L	3997	+4.21	67.4	0.02	3660	+4.21	67.40	0.02	337	+4.11	60.9	0.04	0.015
Serum thyroxine, nmol/L	3997	+4.7	109.9	0.01	3660	+4.70	109.9	0.01	337	+4.63	102.5	0.03	0.12
Serum TSH, mU/L	3925	+0.42	1.52	0.03	3594	+0.42	1.522	0.03	331	+0.41	1.506	0.06	0.94
Serum antimicrosomal Ab, U/mL	3927	−0.61	0.54	0.06	3596	−0.61	0.543	0.06	331	−0.55	0.576	0.18	0.73
Serum anti-thyroglobulin Ab, U/mL	3927	−0.06	0.94	0.04	3596	−0.06	0.942	0.04	331	−0.03	0.970	0.09	0.76
White blood cell count	3998	+1.94	6.96	0.01	3661	+1.93	6.890	0.01	337	+2.01	7.463	0.03	0.015
Lymphocyte percent	3998	+3.46	31.81	0.01	3661	+3.47	32.14	0.01	337	+3.40	29.96	0.03	0.026
Mononuclear percent	3920	+1.67	5.310	0.04	3584	+1.67	5.312	0.04	336	+1.69	5.419	0.05	0.77
Granulocyte percent	3920	+4.10	60.34	0.01	3584	+4.10	60.34	0.01	336	+4.13	62.17	0.02	0.12
Lymphocyte number	3998	+0.80	2.23	0.01	3661	+0.79	2.20	0.01	337	+0.81	2.247	0.03	0.67
Mononuclear number	3905	−0.99	0.370	0.03	3572	−1.00	0.368	0.03	333	−0.91	0.402	0.05	0.039
Granulocyte number	3920	+1.43	4.18	0.02	3584	+1.43	4.18	0.02	336	+1.53	4.618	0.03	0.017
Red blood cell count, SI	3997	+1.55	4.71	0.00	3660	+1.55	4.71	0.00	337	+1.55	4.711	0.00	0.74
Hemoglobin, g/L	3998	+4.96	142.5	0.00	3661	+4.95	141.2	0.00	337	+4.97	144.0	0.01	0.001
Hematocrit, L/L = 1	3997	−0.87	0.420	0.00	3660	−0.87	0.419	0.00	337	−0.85	0.427	0.01	0.001
Mean cell volume, fL	3998	+4.49	89.12	0.00	3661	+4.49	89.12	0.00	337	+4.51	90.92	0.00	<0.001
Mean cell hemoglobin, pg	3997	+3.41	30.3	0.00	3660	+3.41	30.27	0.00	337	+3.43	30.87	0.00	<0.001
Mean cell hemoglobin conc., SI	3997	+5.82	337.0	0.00	3660	+5.82	337.0	0.00	337	+5.83	340.3	0.00	0.30
Red cell distribution width, %	3998	−2.05	0.130	0.00	3661	−2.05	0.129	0.00	337	−2.02	0.132	0.00	<0.001
Platelet count: SI	3998	+5.54	254.7	0.01	3661	+5.53	252.1	0.01	337	+5.54	254.7	0.02	0.87
Platelet distribution width, %	3973	+2.81	16.61	0.00	3640	+2.80	16.44	0.00	333	+2.80	16.44	0.00	0.94
Mean platelet volume, fL	3997	+2.13	8.41	0.00	3660	+2.13	8.41	0.00	337	+2.12	8.331	0.01	0.30
Lead, µmol/L	3999	−2.13	0.118	0.03	3662	−2.15	0.116	0.04	337	−1.82	0.162	0.06	<0.001
Erythrocyte protoporphyrin, SI	3999	−0.19	0.83	0.01	3662	−0.18	0.84	0.01	337	−0.25	0.779	0.03	0.029
Serum iron, µmol/L	4000	+2.71	15.02	0.01	3663	+2.71	15.03	0.01	337	+2.73	15.33	0.04	0.67
Serum TIBC, µmol/L	3997	+4.16	64.07	0.01	3660	+4.15	63.43	0.01	337	+4.18	65.37	0.01	0.067
Serum ferritin, µmol/L	3998	+4.43	83.93	0.03	3661	+4.41	82.27	0.03	337	+4.55	94.63	0.09	0.14
Serum folate, nmol/L	4000	+2.57	13.07	0.04	3663	+2.61	13.60	0.04	337	+2.03	7.614	0.07	<0.001
RBC folate, nmol/L	3952	+6.03	415.7	0.03	3615	+6.05	424.1	0.02	337	+5.73	307.9	0.06	<0.001
Serum vitamin B-12, pmol/L	3999	+5.79	327.0	0.01	3662	+5.81	333.6	0.01	337	+5.52	249.6	0.04	<0.001
Serum vitamin C, nmol/L	3841	+3.50	33.11	0.04	3510	+3.54	34.47	0.04	331	+3.11	22.42	0.09	<0.001
Serum normalized calcium, mmol/L	3709	+0.21	1.233	0.00	3410	+0.21	1.234	0.00	308	+0.21	1.234	0.00	0.55
Serum total calcium, nmol/L	3993	+0.84	2.316	0.00	3657	+0.84	2.316	0.00	336	+0.84	2.314	0.00	0.029
Serum selenium, nmol/L	3977	+0.47	1.599	0.01	3642	+0.47	1.600	0.01	335	+0.51	1.665	0.02	0.004
Serum vitamin A, µmol/L	3993	+0.66	1.934	0.01	3656	+0.66	1.935	0.01	337	+0.67	1.954	0.02	0.57
Serum vitamin E, µmol/L	3993	+3.26	26.05	0.01	3656	+3.27	26.31	0.01	337	+3.16	23.571	0.02	<0.001
Serum alpha carotene, µmol/L	3993	−2.62	0.073	0.028	3622	−2.60	0.074	0.030	322	−2.87	0.057	0.08	0.003
Serum beta carotene, µmol/L	3991	−1.24	0.289	0.02	3656	−1.22	0.295	0.02	335	−1.56	0.210	0.07	<0.001
Serum beta-cryptoxanthin, µmol/L	3991	−1.94	0.143	0.02	3655	−1.92	0.147	0.02	336	−2.17	0.114	0.05	<0.001
Serum lutein/zeaxanthin, µmol/L	3992	−1.07	0.343	0.01	3656	−1.07	0.343	0.01	336	−1.15	0.317	0.03	0.003
Serum lycopene, µmol/L	3990	−0.91	0.402	0.02	3655	−0.90	0.407	0.02	335	−1.00	0.368	0.03	0.012
Serum retinyl esters, µmol/L	3974	−1.78	0.169	0.02	3641	−1.75	0.174	0.02	333	−2.09	0.124	0.05	<0.001
Serum cholesterol, mmol/L	3994	+1.65	5.206	0.01	3657	+1.65	5.207	0.01	337	+1.66	5.259	0.02	0.56
Serum triglycerides, mmol/L	3994	+0.33	1.391	0.02	3657	+0.32	1.377	0.02	337	+0.39	1.476	0.05	0.30
Serum HDL-cholesterol, mmol/L	3972	+0.20	1.221	0.01	3640	+0.21	1.234	0.01	332	+0.17	1.185	0.02	0.12
Serum C-reactive protein, mg/dL	3983	−1.20	0.301	0.02	3646	−1.21	0.298	0.02	337	−1.19	0.304	0.04	0.65
Serum hepatitis A Ab	4000	+0.47	1.600	0.01	3663	+0.47	1.600	0.01	337	+0.48	1.616	0.03	0.67
Serum hepatitis B core Ab	4000	+0.65	1.915	0.00	3663	+0.65	1.916	0.00	337	+0.64	1.896	0.02	0.66
Serum hepatitis C Ab	4000	+0.67	1.954	0.00	3663	+0.68	1.974	0.00	337	+0.67	1.954	0.01	0.46
Serum rubella Ab, IU	3885	+4.31	74.44	0.04	3555	+4.31	74.44	0.04	330	+4.26	70.81	0.13	0.74
Serum sodium, mmol/L	3997	+4.95	141.1	0.00	3641	+4.94	139.77	0.00	336	+4.95	141.1	0.00	0.81
Serum potassium, mmol/L	3977	+1.41	4.095	0.00	3641	+1.41	4.096	0.00	336	+1.40	4.055	0.01	0.30
Serum chloride, mmol/L	3977	+4.64	103.5	0.00	3641	+4.64	103.5	0.00	336	+4.64	103.5	0.00	0.35
Serum bicarbonate, mmol/L	4000	+3.30	27.11	0.01	3663	+3.30	27.11	0.01	337	+3.30	27.11	0.02	0.91
Serum total calcium, mmol/L	3977	+0.83	2.293	0.00	3641	+0.83	2.290	0.00	336	+0.84	2.320	0.00	0.021
Serum phosphorus, mmol/L	3977	+0.09	1.094	0.01	3641	+0.08	1.083	0.01	336	+0.10	1.105	0.01	0.10
Serum uric acid, µmol/L	3977	+5.72	304.9	0.01	3641	+5.71	301.87	0.01	336	+5.82	336.97	0.02	<0.001
Serum glucose, mmol/L	3974	+1.68	5.366	0.01	3639	+1.68	5.366	0.01	335	+1.69	5.419	0.03	0.66
Serum blood urea nitrogen, SI	3977	+1.55	4.711	0.01	3641	+1.55	4.711	0.01	336	+1.54	4.664	0.02	0.49
Serum total bilirubin, µmol/L	3977	+2.25	9.487	0.02	3641	+2.25	9.487	0.02	336	+2.30	9.974	0.04	0.18
Serum creatinine, µmol/L	3977	+4.52	91.83	0.00	3641	+4.52	91.83	0.00	336	+4.61	100.48	0.02	<0.001
Serum iron, µmol/L	3977	+2.66	14.29	0.01	3641	+2.66	14.30	0.01	336	+2.68	14.58	0.04	0.65
Serum cholesterol, mmol/L	3977	+1.68	5.365	0.01	3641	+1.68	5.366	0.01	336	+1.69	5.419	0.02	0.40
Serum triglycerides, mmol/L	3977	+0.29	1.336	0.02	3641	+0.29	1.336	0.02	336	+0.35	1.419	0.06	0.34
Aspartate aminotransferase, U/L	3977	+3.02	20.49	0.01	3641	+3.02	20.49	0.01	336	+3.02	20.49	0.05	0.90
Alanine aminotransferase, U/L	3977	+2.86	17.46	0.02	3641	+2.87	17.63	0.02	336	+2.73	15.33	0.07	0.058
Gamma glutamyl transferase, U/L	3976	+3.17	23.80	0.02	3640	+3.16	23.57	0.02	336	+3.31	27.38	0.07	0.056
Serum lactate dehydrogenase, U/L	3976	+5.10	164.0	0.01	3641	+5.10	164.02	0.01	335	+5.10	164.02	0.01	0.86
Serum alkaline phosphatase, U/L	3977	+4.37	79.04	0.01	3641	+4.36	78.26	0.01	336	+4.52	91.83	0.02	<0.001
Serum total protein, g/L	3977	+4.29	72.96	0.00	3641	+4.29	72.97	0.00	336	+4.29	72.97	0.01	0.80
Serum albumin, g/L	3977	+3.71	40.85	0.00	3641	+3.71	40.85	0.00	336	+3.72	41.26	0.01	0.049
Serum globulin, g/L	3977	+3.46	31.81	0.01	3641	+3.46	31.82	0.01	336	+3.45	31.50	0.01	0.22
Serum osmolality, mmol/kg	3977	+5.64	281.4	0.00	3641	+5.64	281.46	0.00	336	+5.64	281.46	0.00	0.68
Glycated hemoglobin, %	3995	+1.68	5.365	0.01	3658	+1.68	5.365	0.01	337	+1.69	5.419	0.02	0.45
Plasma glucose, mmol/L	3996	+1.67	5.312	0.01	3659	+1.67	5.312	0.01	337	+1.68	5.365	0.02	0.66
Urinary cadmium, nmol/L	3964	+1.21	3.353	0.05	3637	+1.19	3.287	0.05	327	+1.47	4.349	0.10	0.003
Urinary creatinine, mmol/L	3960	+2.18	8.846	0.02	3635	+2.17	8.758	0.02	325	+2.30	9.974	0.06	0.048
Urinary albumin, µg/L	3960	+1.58	4.854	0.06	3635	+1.55	4.711	0.06	325	+1.91	6.753	0.16	0.041
Urinary iodine, µg/L	3956	+2.54	12.67	0.05	3631	+2.54	12.68	0.05	325	+2.51	12.30	0.07	0.68
	***n***	**Mean**	**%**	**SE**	***n***	**Mean**	**%**	**SE**	***n***	**Mean**	**%**	**SE**	P_Hcy_
NHANES 1999–2006	10,151		100.0	0.0	9704		95.9	0.3	447		4.1	0.3	
Age (y)	10,151	45.8		0.20	9704	45.6		0.20	447		50.4	0.6	<0.001
Sex, % men	10,151		48.6	0.4	9704		48.2	0.4	447		59.9	0.4	0.005
Race/ethnicity	7605				7260								0.50
NH white			73.0	2.1			73.1	2.0	345		72.1	4.1	
NH black			10.9	1.1			10.7	1.0			16.5	2.5	
MA			6.8	1.0			7.0	1.0			3.0	0.6	
Other			9.2	1.1			9.3	1.1			8.4	2.6	
Poverty status	9471				9060								<0.001
PIR ≥ 125%			81.2	0.7			84.6	0.7	411		75.1	2.4	
PIR < 125%			15.8	0.7			15.4	0.7			24.9	2.4	
Hcy, Log_e_	10,151	+2.08		0.01	9704	+2.04		0.01	447	+2.94		0.02	<0.001
	***n***	**Mean**	**%**	**SE**	***n***	**Mean**	**%**	**SE**	***n***	**Mean**	**%**	**SE**	P_Hcy_
HANDLS 2004–2018	245		100.0		220		89.8		25		10.2		
Age (y)	245	49.2		0.56	220	48.6		0.59	25	54.3		1.5	0.002
Sex, % men	245		51.0		220		48.6		25		72.0		0.032
Race/ethnicity	245				220				25				
Whites			29.8				31.8				12.0		0.052
AA			70.2				68.2				88.0		
Poverty status	245				220				25				0.069
PIR ≥ 125%			37.1				39.1				20.0		
PIR < 125%			62.9				60.9				80.0		
Hcy, Log_e_	245		+2.26	0.02	220	+2.19		0.02	25	+2.90		0.05	<0.001

Abbreviations: HANDLS = Healthy Aging in Neighborhoods of Diversity Across the Life Span: Hcy = Homocysteine; NH = non-Hispanic; NHANES = National Health and Nutrition Examination Surveys; PIR = Poverty Income Ratio; ROC = Receiver Operating Characteristics analysis. ^1^ All analyses, except for HANDLS, were adjusted for sampling design complexity, to obtain corrected standard errors for means and proportions. Means and proportions of study variables were compared across categories of Hcy (0 = normal, 1 = hyperhomocysteinemic), using simple linear regression for continuous variables and logistic regression in which Hcy category was the outcome for categorical variables. *p* value presented is associated with the regression coefficient. In NHANES, the regression models were also adjusted for sampling design complexity. Biochemical and hematological markers were compared by Hcy categories on their Log_e_-transformed scale (Mean, SE). However, the exponentiated mean is also presented for better clinical interpretation.

**Table 2 nutrients-12-00950-t002:** Selected independent binary correlates and indices (I and II) of elevated homocysteine: Reduced multiple logistic regression models and model-specific area under the ROC curve; pooled NHANES III, phase 2 and 1999–2006 ^1^.

	Elevated Homocysteine		
	OR	95% CI	*p*-Value
Model 1: Binary predictors, (*n* = 14,739)			
Lower serum folate ^2^	3.49	(2.63,4.63)	<0.001
Higher serum creatinine ^3^	1.86	(1.51,2.29)	<0.001
Older age ^4^	1.95	(1.56,2.44)	<0.001
Lower serum vitamin B-12 ^5^	2.52	(1.98,3.21)	<0.001
Higher MCH ^6^	1.60	(1.28,2.02)	<0.001
Higher RDW ^7^	1.24	(1.01,1.54)	0.044
Higher SUA ^8^	1.67	(1.35,2.06)	<0.001
Higher alkaline phosphatase ^9^	1.71	(1.35,2.15)	<0.001
Higher serum cotinine ^10^	1.77	(1.44,2.17)	<0.001
Model 2: Index I ≥ 5 (*n* = 14,739)	7.43	(5.75,9.61)	<0.001
Model 3: Index II ≥ 5, (*n* = 14,829)	6.90	(5.37,8.84)	<0.001

Abbreviations: HANDLS = Healthy Aging in Neighborhoods of Diversity Across the Life Span: Hcy = Homocysteine; NHANES = National Health and Nutrition Examination Surveys; ROC = Receiver Operating Characteristics analysis. ^1^ All elements of Indices I and II were Log_e_ transformed. Cut-points are determined using ROC analysis, using highest sensitivity/specificity combinations. Binary components are entered simultaneously in Model 1. Model 2 includes Index I which sums binary variables: “lower serum folate” (1 = yes, 0 = no), “Higher serum creatinine”, “Older age”, “Lower serum vitamin B-12”, “Higher MCH”, “Higher RDW”, “Higher SUA”, “Higher alkaline phosphatase” and “Higher serum cotinine”, with a possible range of 0–9. Model 3 included Index II which sums binary variables of Index I excluding “Higher blood lead” and “Higher serum cotinine”. Possible range: 0–7. Cut-points for Indices I and II were also determined using ROC curve analysis, with an optimal cut-point selected using the highest sensitivity/specificity combination. ^2^ Serum folate, Log_e_, in nmol/L, < 2.83; ^3^ Serum creatinine, Log_e_, in µmol/L ≥4.481; ^4^ Older age, in years, ≥49, ^5^ Serum vitamin B-12, Log_e_, in pmol/L, <5.74; ^6^ Mean cell hemoglobin, Log_e_, in pg, ≥3.422; ^7^ Red cell distribution width, Log_e_, in %, ≥2.553; ^8^ Serum Uric Acid, Log_e_, in µmol/L, ≥5.826; ^9^ Serum alkaline phosphatase, Log_e_, in µmol/L, ≥4.356 U/L; ^10^ Serum cotinine, Log_e_, in ng/mL, −0.579.

**Table 3 nutrients-12-00950-t003:** Baseline serum homocysteine as a predictor of selected biochemical and hematological parameters at baseline and their change over time: mixed-effects linear regression models; HANDLS 2004–2018 ^1^.

Outcome	Intercept		Time		Hcy		Hcy × Time		(*n*) k
	γ_00_ ± SE	*p*	γ_10_ ± SE	*p*	γ_0a_ ± SE	*p*	γ_1a_ ± SE	*p*	
Serum folate, nmol/L									
Model 1: Age-adjusted	+33.6 ± 1.2	<0.001	+0.11 ± 0.16	0.47	−0.21 ± 0.32	0.51	−0.05 ± 0.05	0.36	(243) k = 2.4
Model 2: Socio-demographic adjusted	+29.8 ± 6.0	<0.001	+2.07 ± 0.83	0.012	−0.25 ± 0.34	0.45	−0.05 ± 0.05	0.37	(243) k = 2.4
Model 3: Multivariable-adjusted	+35.1 ± 6.2	<0.001	+1.39 ± 0.87	0.11	−1.41 ± 0.43	0.001	+0.03 ± 0.07	0.69	(227) k = 2.4
Serum creatinine, µmol/L									
Model 1: Age-adjusted	+107.9 ± 7.1	<0.001	−1.18 ± 0.30	<0.001	+18.3 ± 1.89	<0.001	+0.11 ± 0.10	0.24	(243) k = 2.4
Model 2: Socio-demographic adjusted	+100.1 ± 37.1	0.007	−3.40 ± 1.50	0.024	+18.3 ± 2.00	<0.001	+0.14 ± 0.10	0.16	(243) k = 2.4
Model 3: Multivariable-adjusted	+117.7 ± 35.1	0.001	−3.93 ± 1.60	0.014	+19.8 ± 2.01	<0.001	+0.24 ± 0.12	0.045	(227) k = 2.4
Serum vitamin B-12, µmol/L									
Model 1: Age-adjusted	+472 ± 13	<0.001	−11.2 ± 1.6	<0.001	−4.64 ± 3.69	0.20	+0.85 ± 0.53	0.11	(243) k = 2.4
Model 2: Socio-demographic adjusted	+334 ± 68	<0.001	+6.1 ± 8.6	0.48	−5.19 ± 3.80	0.17	+0.87 ± 0.55	0.11	(243) k = 2.4
Model 3: Multivariable-adjusted	+307 ± 69	<0.001	−8.0 ± 8.9	0.37	−11.0 ± 4.9	0.023	+1.66 ± 0.69	0.015	(227) k = 2.4
Mean cell hemoglobin, pg									
Model 1: Age-adjusted	+29.6 ± 0.2	<0.001	−0.003 ± 0.002	0.56	+0.03 ± 0.05	0.53	−0.003 ± 0.005	0.56	(244) k = 2.4
Model 2: Socio-demographic adjusted	+30.9 ± 0.9	<0.001	−0.04 ± 0.10	0.67	+0.02 ± 0.05	0.72	−0.004 ± 0.005	0.49	(244) k = 2.4
Model 3: Multivariable-adjusted	+29.2 ± 0.97	<0.001	+0.03 ± 0.10	0.78	+0.04 ± 0.06	0.50	+0.01 ± 0.01	0.42	(227) k = 2.4
Red cell distribution width, %									
Model 1: Age-adjusted	+13.8 ± 0.11	<0.001	+0.12 ± 0.01	<0.001	*+0.05 ± 0.03*	*0.079*	+0.007 ± 0.003	0.035	(244) k = 2.4
Model 2: Socio-demographic adjusted	+12.7 ± 0.57	<0.001	+0.18 ± 0.06	0.002	+0.04 ± 0.03	0.16	+0.008 ± 0.004	0.021	(244) k = 2.4
Model 3: Multivariable-adjusted	+12.7 ± 0.5	<0.001	+0.19 ± 0.06	0.002	−0.04 ± 0.04	0.20	*+0.009 ± 0.005*	*0.056*	(227) k = 2.4
Serum uric acid, µmol/L									
Model 1: Age-adjusted	+314.8 ± 5.3	<0.001	+3.79 ± 0.7	<0.001	+7.48 ± 1.45	<0.001	−0.20 ± 0.22	0.36	(243) k = 2.4
Model 2: Socio-demographic adjusted	+226.8 ± 26.7	<0.001	+2.80 ± 3.51	0.42	+5.30 ± 1.48	<0.001	−0.12 ± 0.23	0.62	(243) k = 2.4
Model 3: Multivariable-adjusted	+228.6 ± 27.0	<0.001	+2.01 ± 3.61	0.58	+8.11 ± 1.88	<0.001	−0.01 ± 0.29	0.97	(227) k = 2.4
Serum alkaline phosphatase, U/L									
Model 1: Age-adjusted	+90.8 ± 2.1	<0.001	−0.88 ± 0.22	<0.001	+2.66 ± 0.56	<0.001	−0.03 ± 0.07	0.70	(243) k = 2.4
Model 2: Socio-demographic adjusted	+70.7 ± 10.5	<0.001	−0.39 ± 1.19	0.74	+2.54 ± 0.58	<0.001	−0.002 ± 0.08	0.98	(243) k = 2.4
Model 3: Multivariable-adjusted	+75.2 ± 10.7	<0.001	−0.27 ± 1.23	0.83	+2.72 ± 0.74	<0.001	*+0.16 ± 0.10*	*0.095*	(227) k = 2.4

Abbreviations: HANDLS = Healthy Aging in Neighborhoods of Diversity Across the Life Span; Hcy = Homocysteine; k = mean number of observations/participant; *n* = Number of participants. ^1^ All selected biochemical and hematological markers were measured in SI units. Measures were not Log_e_ transformed. Model 1 was adjusted for age, centered at 48.8 years. Model 2 was additionally adjusted for sex, race (African Americans vs. Whites), and poverty status (above vs. below poverty). Model 3 was additionally adjusted for all remaining biochemical and hematological measures that were selected. In Model 3, Folate was centered at 32.4, creatinine at 105.8, vitamin B-12 at 494.76, Mean cell hemoglobin at 29.46, red cell distribution width at 13.64, serum uric acid at 306.2, and alkaline phosphatase at 82.

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
