# Peer review of "Biochemical and Hematological Correlates of Elevated Homocysteine in National Surveys and a Longitudinal Study of Urban Adults"

_nutrients, 2020, doi:10.3390/nu12040950_

Round 1

Reviewer 1 Report

 Unfortunately this study is not related to my expertise.

Author Response

Response: We appreciate your efforts. Thank you!

Reviewer 2 Report

This is a very well-written manuscript. However, I have some concerns as follows.

MTHFR polymorphism is a strong risk factor for increased homocysteine level. Howevert, this study did not include MTHFR genotypes, which is one of the most important determinants.

The authors did not mention anything about specimen handling issue, which is cruicial to get accurate results from homocysteine assay.

I don't fully understand the clinical impact of this analysis to predict serum homocysteine level, using biochemical and hematological vaules and without well-known variables.
For instance, serum cotinine and urinary cadmium are related to only recent smoking status. Macrocytic anemia might be the result of cobalamine deficiency or other causes, not the predictor of serum homocysteine level.  

Author Response

Reviewer 2:

Comments and Suggestions for Authors

1) This is a very well-written manuscript. However, I have some concerns as follows.

Response: Thank you!

2) MTHFR polymorphism is a strong risk factor for increased homocysteine level. However, this study did not include MTHFR genotypes, which is one of the most important determinants.

Response: We appreciate this comment. The purpose of this study was to uncover biochemical and hematological markers associated with Hcy rather than genetic and lifestyle factors. Thus, we only included a clarification in the introduction as to what are the factors that generally determine Hcy rather than including it among limitations. The introduction was modified as follows:

“During One Carbon metabolism (OCM) cycles, one key enzymatic reaction involves…”

“Serum folate, vitamins B-6 and B-12 levels were strong inverse predictors of elevated blood Hcy [1-6], as are genetic polymorphisms associated with the OCM, such as MTHFR C667T, associated with reduced Methylene tetrahydrofolate reductase (MTHFR) enzymatic activity[7].”

3) The authors did not mention anything about specimen handling issue, which is crucial to get accurate results from homocysteine assay.

Response: The following was added: “NHANES specimen storage were consistent across waves. Upon arrival at CDC or contract laboratories, the frozen specimens were sorted by vial type, and stored initially at -20 C. The refrigerated samples were stored at 4-8 C. Frozen specimens whose analysis may be delayed were stored at -70 C or lower[29].”

4) I don't fully understand the clinical impact of this analysis to predict serum homocysteine level, using biochemical and hematological values and without well-known variables.

Response: We appreciate this comment. The fixed variables included in all LASSO models were socio-demographic and SES variables. All others were selected out of all possible variable combinations using machine learning techniques with cross-validation, minBIC and adaptive LASSO. The main implications are as follows: 1) To discover the most predictive measures of elevated Hcy out of >80 markers; 2) Create indices that can be used as surrogates of elevated Hcy in studies that did not measure Hcy per se; 3) Enhance understanding as to why elevated Hcy may increase the risk of certain age-related diseases including AD, by discovering new markers that are highly predictive of elevated Hcy.

These points are now better clarified in the Introduction as follows:

“Generally, there is paucity of research in the following areas: 1) Discovering the most predictive measures of elevated Hcy out of selected biochemical and hematological markers; 2) Creating indices that can be used as surrogates of elevated Hcy in studies that did not measure Hcy per se; 3) Enhancing understanding as to why elevated Hcy may increase the risk of certain age-related diseases including AD, by discovering novel markers that are highly predictive of elevated Hcy.”

5) For instance, serum cotinine and urinary cadmium are related to only recent smoking status.

Response: This was added to the Discussion as follows: “It is worth noting that both serum cotinine and urinary cadmium were linked to recent smoking [60,61].”

6) Macrocytic anemia might be the result of cobalamine deficiency or other causes, not the predictor of serum homocysteine level.  

Response: We appreciate this comment. The following was added to the Discussion: “Thus, Hcy may well be a correlate of higher MCH, resulting from any or both vitamin deficiencies; and a higher Hcy may result in higher MCH over time. Nevertheless, our findings indicated that although an independent correlate of elevated Hcy, MCH was not among the strongest predictors based on ROC analyses. Furthermore, our longitudinal analyses did not indicate that baseline Hcy was associated with faster increase in MCH over time. Nevertheless, larger studies are needed to corroborate these findings.”

Reviewer 3 Report

To the authors,

The paper of Beydoun et al describes a set of 9 predictors of serum homocysteine by applying machine learning tools in combination with ROC analysis.

I have the following comments;

  • The word cotinine is mispelled in the abstract. Please adapt
  • The authors describe a set of 9 independent predictors including folate, vitamin B12, H band RDW. This is interestingly as these 4 four parameters are highly related in particular in case of folate- and vitamin B12 deficiency. I think the paper would merit if the authors discuss how these 4 parameters are independently associated with serum homocysteine.
  • The authors discuss about difference in assays to measure homocysteine. However, if within the same cohorts two methods have been used I would suggest to use a factor to recalculate the values in such as way that the general cut-off of >14 uM would be applicable. Or at least the authors should add reference intervals for methods used to compare methods.
  • In figure 2 the authors have give a ROC plot. However, a ROC plot has sensitivity on the y-axis and 1-specificity on the x-axis. The authors have now plotted specificiy on the x-asis, which I would suggest to change into 1-specificity.
  • Table I is very comprehensive and is therefore less informative. I would suggest to put in into supplements instead.
  • In figure 8 and 9 two ROC plots are shown. I would suggest to make one figure with two lines in the same ROC plot so that you can easily compare both ROC plots.
  • In table I, it can be seen that folate is the strongest predictor of serum homocysteine. The authors could discuss this more thorougly in the discussion of the paper.

Author Response

Reviewer 3:

Comments and Suggestions for Authors

To the authors,

The paper of Beydoun et al describes a set of 9 predictors of serum homocysteine by applying machine learning tools in combination with ROC analysis.

I have the following comments;

Response: Thank you!.

  1. The word cotinine is mispelled in the abstract. Please adapt

Response: Done.

  1. The authors describe a set of 9 independent predictors including folate, vitamin B12, H band RDW. This is interestingly as these 4 parameters are highly related in particular in case of folate- and vitamin B12 deficiency. I think the paper would merit if the authors discuss how these 4 parameters are independently associated with serum homocysteine.

Response: The following was added to the first paragraph of the Discussion: “The study uncovered up to 9 independent predictors for elevated Hcy, some of which have been found to be correlated with each other (e.g. serum folate, B-12, MCH, and RDW) in previous studies as well as with hyperhomocysteinemia.”

  1. The authors discuss about difference in assays to measure homocysteine. However, if within the same cohorts two methods have been used I would suggest to use a factor to recalculate the values in such as way that the general cut-off of >14 uM would be applicable. Or at least the authors should add reference intervals for methods used to compare methods.

Response: Two different methods were used in NHANES III and NHANES 1999-2006. There were no repeats, since those are cross-sectional data. Thus, we were not able to compare values within participants using different Hcy methods. This was added among limitations: “Attempts to calibrate those two methods are needed in future studies with repeat measures using both methods.”

  1. In figure 2 the authors have given a ROC plot. However, a ROC plot has sensitivity on the y-axis and 1-specificity on the x-axis. The authors have now plotted specificiy on the x-asis, which I would suggest to change into 1-specificity.Response: Done. All Figures are now presented as sensitivity vs. 1-specificity. We appreciate this comment.
  2.  
  3.  
  4. Table I is very comprehensive and is therefore less informative. I would suggest to put in into supplements instead.   
  5. The following was added for clarification: “It is worth noting that those associations are crude, not adjusted for socio-demographic factors such as age and sex.”
  6. Response: We appreciate this comment. However, we feel that Table I is important for the reader to go through all the unadjusted associations between normal and elevated Hcy groups, without having to refer to the supplement. The supplement, on the other hand, describes the methods used to measure those biomarkers in each cohort.
  7.  
  8. In figure 8 and 9 two ROC plots are shown. I would suggest to make one figure with two lines in the same ROC plot so that you can easily compare both ROC plots.

Response: We would have liked to do so, however, those would largely overlap and we would not be able to separate them adequately for the reader to recognize each one.

  1. In table I, it can be seen that folate is the strongest predictor of serum homocysteine. The authors could discuss this more thoroughly in the discussion of the paper.Response: The following was added to the Results section: “Among the known predictive factors, serum folate was shown to have an unadjusted mean of 7.6 in the elevated Hcy group compared to 13.6 in the normal group, suggesting a strong inverse relationship.”And in the Discussion section: “Both adjusted and unadjusted associations indicated that serum folate was the most predictive factor that was inversely related to elevated Hcy.”  
  2. Reference:
  3.  
  4.  
  5.  
  1. Hatzis, C.M.; Bertsias, G.K.; Linardakis, M.; Scott, J.M.; Kafatos, A.G. Dietary and other lifestyle correlates of serum folate concentrations in a healthy adult population in Crete, Greece: a cross-sectional study. Nutr J 2006, 5, 5, doi:10.1186/1475-2891-5-5.
  2. Manavifar, L.; Nemati Karimooy, H.; Jamali, J.; Talebi Doluee, M.; Shirdel, A.; Nejat Shokohi, A.; Fatemi Nayyeri, M. Homocysteine, Cobalamin and Folate Status and their Relations to Neurocognitive and Psychological Markers in Elderly in Northeasten of Iran. Iran J Basic Med Sci 2013, 16, 772-780.
  3. Song, J.H.; Park, M.H.; Han, C.; Jo, S.A.; Ahn, K. Serum Homocysteine and Folate Levels are Associated With Late-life Dementia in a Korean Population. Osong Public Health Res Perspect 2010, 1, 17-22, doi:10.1016/j.phrp.2010.12.006.
  4. Yang, X.; Gao, F.; Liu, Y. Association of homocysteine with immunological-inflammatory and metabolic laboratory markers and factors in relation to hyperhomocysteinaemia in rheumatoid arthritis. Clin Exp Rheumatol 2015, 33, 900-903.
  5. Cheng, C.H.; Huang, Y.C.; Chen, F.P.; Chou, M.C.; Tsai, T.P. B-vitamins, homocysteine and gene polymorphism in adults with fasting or post-methionine loading hyperhomocysteinemia. Eur J Nutr 2008, 47, 491-498, doi:10.1007/s00394-008-0752-5.
  6. Selhub, J.; Jacques, P.F.; Wilson, P.W.; Rush, D.; Rosenberg, I.H. Vitamin status and intake as primary determinants of homocysteinemia in an elderly population. JAMA 1993, 270, 2693-2698, doi:10.1001/jama.1993.03510220049033.
  7. Rozen, R. Genetic predisposition to hyperhomocysteinemia: deficiency of methylenetetrahydrofolate reductase (MTHFR). Thromb Haemost 1997, 78, 523-526.